# Paraneoplastic Resolution Holds Prognostic Utility in Patients with Metastatic Renal Cell Carcinoma

**DOI:** 10.3390/cancers16213678

**Published:** 2024-10-30

**Authors:** Gregory Palmateer, Edouard H. Nicaise, Taylor Goodstein, Benjamin N. Schmeusser, Dattatraya Patil, Nahar Imtiaz, Daniel D. Shapiro, Edwin J. Abel, Shreyas Joshi, Vikram Narayan, Kenneth Ogan, Viraj A. Master

**Affiliations:** 1Department of Urology, Emory University School of Medicine, Atlanta, GA 30322, USA; gpalmat@emory.edu (G.P.); enicais@emory.edu (E.H.N.); taylor.alexandra.goodstein@emory.edu (T.G.); dattatraya.patil@emory.edu (D.P.); shamsunnahar.imtiaz@emory.edu (N.I.); shreyas.joshi@emory.edu (S.J.); vikram.narayan@emory.edu (V.N.); kogan@emory.edu (K.O.); 2Department of Urology, University of Indiana School of Medicine, Indianapolis, IN 46202, USA; bschmeus@iu.edu; 3Department of Urology, University of Wisconsin School of Medicine and Public Health, Madison, WI 53726, USA; ddshapiro@wisc.edu (D.D.S.); abel@urology.wisc.edu (E.J.A.); 4Winship Cancer Institute, Emory University School of Medicine, Atlanta, GA 30322, USA

**Keywords:** paraneoplastic syndromes, metastatic RCC, IMDC, MSKCC, SCREEN, survival

## Abstract

This is the first comprehensive study to examine the impact of paraneoplastic syndrome (PNS) resolution on overall and cancer-specific survival in patients with metastatic renal cell carcinoma (mRCC). PNS can manifest as hematologic, metabolic, immunologic, or constitutional abnormalities, and their presence prior to surgery is associated with worse survival. The PNS we assessed were those previously described by Moldovan et al., including neutrophil–lymphocyte. We found that resolution of one or more PNS by one year after surgery was independently associated with improved overall survival and cancer-specific survival. Furthermore, patients categorized as favorable risk in current mRCC risk models (SCREEN, IMDC, MSKCC) did not experience PNS resolution more often than patients categorized as poor risk. This suggests that PNS resolution may provide distinct prognostic information to better risk stratify patients.

## 1. Introduction

Paraneoplastic syndromes (PNS) encompass a wide variety of clinical signs and symptoms distinct from those caused directly by a primary or metastatic malignancy. These syndromes manifest due to cytokines released from either the tumor or the immune system [1,2]. PNS can manifest as hematologic (anemia, polycythemia, thrombocythemia, thrombocytosis), metabolic (hypercalcemia, hyperuricemia, hepatic dysfunction), immunologic (elevated C-reactive protein [CRP], erythrocyte sedimentation rate [ESR], neutrophil–lymphocyte ratio [NLR]), or constitutional (fever, weight loss, night sweats, hypertension) abnormalities. Renal cell carcinoma (RCC) is well known to be associated with PNS, with 10–54% of patients presenting with at least one laboratory abnormality consistent with PNS at the time of diagnosis [3,4,5,6]. Furthermore, the presence of multiple PNS at presentation has been associated with higher disease stages, worse overall survival (OS), and worse cancer-specific survival (CSS) compared to a lack of PNS at presentation [3,4,5,7]. There is a paucity of literature examining whether resolution of PNS following nephrectomy is associated with improved survival in patients with RCC.

Despite the emergence of several new treatment regimens, only 8–12% of patients with metastatic RCC (mRCC) survive to 5 years [8,9]. Currently, combination systemic therapy utilizing immune checkpoint inhibitors (ICI) is standard of care for patients with metastatic disease [10,11,12]. For well-selected patients who can tolerate major surgery, the combination of cytoreductive nephrectomy followed by adjuvant immunotherapy provides a survival benefit greater than immunotherapy alone [13,14]. Metastatic RCC models such as the Selection for Cytoreductive Nephrectomy (SCREEN), International Metastatic RCC Database Consortium (IMDC), and Memorial Sloan Kettering Cancer Center (MSKCC/Motzer) nomograms can assist clinicians with patient risk stratification and treatment selection. These models utilize laboratory abnormalities commonly observed in PNS to help predict patient prognosis. We sought to determine whether patients with metastatic RCC who have resolution of at least one preoperative PNS within one year following cytoreductive nephrectomy have improved CSS or OS compared to patients without PNS resolution. Additionally, we examined if better risk stratification with current mRCC models is associated with PNS resolution at one year. 

## 2. Materials and Methods

### 2.1. Cohort Selection 

We retrospectively reviewed the prospectively maintained Emory Institutional nephrectomy database for patients with any histology, mRCC, who underwent nephrectomy from 2000 to 2022 and had at least six months of follow-up (Appendix A). Patients with one or more PNS prior to surgery and necessary laboratory tests available 90 days before and 30 days to one year after surgery were included. For patients with multiple preoperative labs available, the values closest to the day of surgery were utilized. Several laboratory values were used to define PNS. These included anemia (hemoglobin < 11.0 g/dL or hematocrit < 33%), thrombocytopenia (platelet count < 100/nL), thrombocytosis (platelet count > 400/nL), hepatic dysfunction (defined by either alanine aminotransferase [ALT] > 50 U/L or aspartate aminotransferase [AST] > 50 U/L), hyperuricemia (males: >7 mg/dL; female: >5.7 mg/dL), hypercalcemia (corrected serum calcium > 10.4 mg/dL), elevated CRP (>5 mg/L), elevated ESR (male: >22 mm/h; female: >29 mm/h), and elevated neutrophil–lymphocyte ratio ([NLR] > 4:1). These were the 10 most common highlighted PNS described by Moldovan et al., including NLR (Appendix A) [4]. Each abnormality present was counted as a separate PNS. Our primary exposure of interest was the count of resolved PNS abnormalities one year after surgery. Resolution was defined as an abnormal value compared to established laboratory cutoff within one year from surgery. If multiple postoperative lab measurements were present, those closest to one year following surgery were used to assess for resolution. 

Patient demographic information including age, sex, race, smoking history, obesity (Body Mass Index [BMI] > 30 kg/m^2^), hypertension, Charlson comorbidity index (CCI), Eastern Cooperative Oncology Group (ECOG) performance status (PS), and American Society of Anesthesiologists (ASA) score were collected. Additionally, we recorded oncologic information such as tumor stage, grade, histology, presence of necrosis, presence of inferior vena cava (IVC) thrombus, and nodal status. All patients were followed until death or data collection cutoff in July 2024. Patients with no further follow-up were censored at the time of their final clinic visit. Our primary outcomes of interest were 10-year OS and CSS. Our secondary outcome of interest was determining whether PNS resolution correlated with lower risk preoperative SCREEN, MSKCC, and IMDC scoring. OS was defined as death attributed to any cause, while CSS was defined as death secondary to renal malignancy. Mortality information was cross-referenced using the electronic medical record, the state of Georgia death registry, and the national (United States) death index. 

### 2.2. Statistical Analysis 

Continuous variables were calculated as the median values with interquartile ranges (IQR), while categorical variables were reported as the total number with percentages. If any missing data for cardinal variables (to be included in the models) were found, then the relevant observation was excluded from analysis altogether. ANOVA and chi-square tests were used to compare differences between groups. Kaplan–Meier curves were utilized to estimate 10-year OS and CSS rates. Multivariable Cox proportional hazards models were constructed to measure the association between PNS resolution at one year with OS and CSS. Harrell’s concordance statistic estimate was calculated and collinearity and interaction was assessed in each model. To identify which individual laboratory abnormalities were associated with worse OS and CSS, an univariable Cox proportional hazards model was utilized. All statistical tests were two-sided with type 1 error set at 0.05. All analyses were performed using SAS version 9.4 (Cary, NC, USA). 

## 3. Results

### 3.1. Cohort Pathoclinical Characteristics

Of the 253 patients who met inclusion criteria, resolution of at least one PNS was observed in 177 (70.0%) within one year. The distribution of preoperative PNS is displayed in Table 1. The most frequent preoperative lab abnormalities were elevated CRP (88.4%), elevated ESR (77.1%), and low hemoglobin (66.3%). Polycythemia, hyperuricemia, and thrombocytopenia were the least common abnormalities, present in only 1, 3, and 6 patients, respectively. Median follow-up time for the cohort was 23.1 (IQR, 8.2–45.6) months. Patients with ≥1 PNS resolution were more likely to have an IVC thrombus (47.5% vs. 27.6%, *p* = 0.003), a larger tumor (median size, 11.2 vs. 9.6 cm, *p* < 0.001), and worse IMDC score (poor, 53.1% vs. 38.2%, *p* = 0.029) compared to patients with no PNS resolution (Table 2). There were no other patient or oncologic characteristics that differed between patients with and without PNS resolution. Resolution of low hematocrit at one year was significantly associated with a poor SCREEN score (*p* = 0.017); however, no other resolved labs were significantly associated with SCREEN, IMDC, and MSKCC scoring (Table 3). 

### 3.2. PNS Resolution and Survival 

OS and CSS rates in the entire study cohort were 29.5% and 42.8% at five years and 15.6% and 29.1% at ten years, respectively. After dividing patients according to PNS resolution, estimated 5- and 10-year OS rates were 15.7% and 11.7% for patients with no PNS resolution, 24.5% and 6.4% for patients with 1 PNS resolution, and 43.0% and 27.6% for patients with ≥2 PNS resolution within one year, respectively (Figure 1). Estimated 5- and 10-year CSS rates were both 36.2% for patients with no PNS resolution, 31.6% and 18.3% for patients with 1 PNS resolution, and 58.2% and 42.8% for patients with ≥2 PNS resolution within one year, respectively (Figure 2). On multivariable analysis, compared to ≥2 PNS resolved at one year, no PNS resolved and 1 PNS resolved were independently associated with worse OS (HR 2.75, *p* < 0.001; HR 1.76, *p* = 0.004) and CSS (HR 2.62, *p* < 0.001; HR 2.24, *p* < 0.001), respectively (Table 4). Other factors independently associated with worse OS were ECOG ≥ 1 (HR 1.66, *p* = 0.005), an intermediate risk SCREEN score (HR 2.04, *p* = 0.046), and poor risk scores on SCREEN (HR 2.99, *p* = 0.003), IMDC (HR 1.56, *p* = 0.043), and MSKCC (HR 1.78, *p* = 0.031). All other variables and their associations to OS and CSS are summarized in Table 4. 

### 3.3. Individual PNS Abnormalities Associated with OS and CSS

Univariable Cox hazards analysis measuring the association between resolution of the examined labs with OS and CSS is demonstrated in Table 5. A lack of resolution in anemia (Hgb HR 3.10; Hct HR 2.53; *p* < 0.001), hypercalcemia (HR 3.98, *p* < 0.001), elevated CRP (HR 2.11, *p* < 0.001), elevated ESR (HR 3.01, *p* < 0.001), elevated LFTs (HR 1.92, *p* = 0.014), and elevated NLR (HR 1.80, *p* = 0.016) were associated with significantly worse OS. Significantly worse CSS was observed in patients with a lack of resolution in anemia (Hgb HR 3.34; Hct HR 3.14; *p* < 0.001), hypercalcemia (HR 5.16, *p* < 0.001), elevated CRP (HR 2.58, *p* < 0.001), elevated ESR (HR 3.31, *p* < 0.001), and elevated LFTs (HR 2.12, *p* = 0.018). 

## 4. Discussion

Lack of resolution of laboratory abnormalities consistent with PNS within one year of cytoreductive nephrectomy is associated with worse 5- and 10-year OS and CSS. After adjusting for potential confounding variables, lack of PNS resolution remained independently associated with worse OS and CSS. On further analysis, a lack of resolution of preoperative anemia, hypercalcemia, elevated CRP, elevated ESR, or elevated LFTs were all independently associated with worse OS and CSS. While scoring poorly on SCREEN, IMDC, or MSKCC was associated with worse OS, risk stratification by these models was not associated with resolution of preoperative PNS at one year. 

### 4.1. Resolution of PNS Abnormalities and Survival

The 5-year OS rate observed in our overall cohort (29.5%) is higher than what has been reported in the literature for mRCC [8,9,10]. Undoubtedly, there is a degree of selection bias within our cohort. Our study only included patients who had undergone cytoreductive nephrectomy; thus, patients who were either poor risk oncologically or poor surgical candidates were excluded. Additionally, in well-selected patients, cytoreductive nephrectomy could independently confer a survival benefit, though modern studies supporting this are currently retrospective in nature and clinical trials are underway [13,15,16,17,18,19]. Also, based on our exclusion criteria, our study did not capture any patients who may have died within six months of nephrectomy. While the prognostic value of preoperative PNS in patients with RCC has been studied, literature examining the prognostic value of PNS resolution following surgery is limited. What little exists has focused mostly on inflammatory markers, such as CRP, ESR, and NLR for predicting survival and response to systemic treatments. Normalization of CRP values following cytoreductive nephrectomy or systemic immunotherapy has been associated with better progression-free survival (PFS) and OS [20,21,22,23,24,25]. The strength of these associations, however, appears dependent on how CRP resolution is defined. Hoeh et al. compared CRP resolution within three months as defined by Ishihara et al. (≤1 mg/dL) and Fukuda et al. (≥30% decrease from baseline) and found CRP resolution to be associated with PFS and OS using CRP ≤ 1 mg/dL and OS only using a ≥30% decrease in CRP. Similarly, we found, using an absolute threshold of CRP ≤ 5 mg/dL, that patients with CRP resolution at one year had better OS and CSS. Likewise, we found that patients with preoperative ESR levels below gender-specific thresholds had better OS and CSS. This aligns with findings previously published by Zhang et al., who found that patients with ESR levels that decreased by >50% from baseline within three months following treatment with sorafenib had improved PFS [26]. Similarly, several studies have found that patients with a reduction in NLR from baseline following immunotherapy had improved OS and PFS [27,28,29]. In our study, NLR resolution was broadly associated with improved OS in patients who received cytoreductive nephrectomy. Certainly, a number of our patients received immunotherapy; however, PD-1/PD-L1 inhibitors have only been available for the last decade and one-third of our cohort had no record of receiving systemic therapy. Thus, NLR resolution may have utility outside of immunotherapy, but focused studies are required. 

The presence of anemia, hyperuricemia, hypercalcemia, thrombocytosis, and elevated LFTs in patients with RCC is associated with worse OS [30,31,32,33,34,35,36,37,38]. It is postulated that thrombocytopenia and polycythemia are also associated with worse prognosis; however, evidence for this in RCC has been limited to case reports [39,40,41,42]. In our study, resolution of preoperative anemia, hypercalcemia, and elevated LFTs were all associated with improved OS and CSS. It is undetermined whether medically altering these lab abnormalities will reproduce the survival benefit observed. Recent investigative efforts looking at the effect of iron optimization and supplementation on survival in patients with cancers in general have provided mixed results. Keding et al. found improved two-year OS across several different malignancies after implementing system-wide iron conservation practices [43]. Conversely, intravenous iron therapy in patients with colorectal cancer was not found to have a significant effect on DFS and OS in a study by Wilson et al. [44]. Additional research is needed to elicit if there is a clinical benefit to optimizing blood iron levels and other laboratory values in RCC.

### 4.2. Resolution of PNS Abnormalities and Current mRCC Models

While scoring poorly on IMDC, MSKCC, and SCREEN models was associated with worse OS, risk stratification as determined by these models was not associated with PNS resolution. Over 50% of IMDC poor-risk patients had at least one PNS resolution, but granular examination of the specific laboratory values did not identify any one lab resolution associated with poor-risk patients. Patients with preoperative PNS resolution are more likely to have improved OS and CSS irrespective of their mRCC risk stratification. By tracking postoperative lab values, clinicians can provide supplemental risk counseling to patients. 

### 4.3. Additional Factors Independently Associated with OS and CSS

An increased ECOG score was also associated with worse OS and CSS in the multivariable Cox hazards model. This corroborates findings from previous studies which have demonstrated patients with mRCC and worse performance status (ECOG ≥ 2) have significantly worse OS [45]. On the other hand, obesity, defined as a BMI ≥ 30 kg/m^2^, was associated with improved CSS. The “obesity paradox” is a well-described phenomenon in RCC, in which patients with obesity have improved CSS compared to patients with a lower BMI [46,47]. Critics have postulated, however, that weight loss, and not low BMI itself, is responsible for worse survival in these lower weight patients with RCC [48]. Surprisingly, an increased CCI score was independently associated with improved CSS and OS on multivariable analysis. This conflicts with current scientific understanding, which is that possessing a greater number of comorbidities is correlated with reduced survival [49]. This finding is likely the result of selection bias. Patients with a greater number of comorbidities are more likely to be followed by clinicians more frequently than those with less comorbidities. Furthermore, patients with a lower CCI score (CCI 0–3) constituted only 15% of our mRCC cohort, which increases the probability this association is by chance alone. 

### 4.4. Strengths, Limitations, and Future Directions

Studies examining PNS in RCC, to date, have primarily measured their preoperative incidence and respective prognosis. Our study herein represents the largest and most comprehensive examination of PNS resolution within a racially diverse cohort of patients with mRCC. We demonstrated that resolution of PNS at one year was independently associated with improved CSS and OS. Importantly, these associations appear to be distinct from current mRCC models, which suggests that they may provide unique prognostic value. There are limitations to our study, however. Firstly, it is a retrospective single-institution study. We were unable to control for when patients had labs drawn. Additionally, by utilizing the lab closest to one year after surgery, additional values falling before it are not given weight. Laboratory equipment and methods may have differed over the last two decades potentially resulting in variation in lab values. We also were unable to standardize the type, timing, and duration of systemic therapy received by patients. As the only National Cancer Institute (NCI)-designated center in the state of Georgia, Emory hospital system serves patients over a large geographic area. Consequently, many patients choose to receive systemic therapy closer to home and from where we do not have access to records of their continued care. By utilizing absolute thresholds for PNS resolution, we were unable to capture proportional changes in PNS biomarkers that did not fall below our predefined established threshold for normal. Finally, by not utilizing a Bonferroni correction, we reduced the possibility of missing true associations in our novel study; however, by doing so, there is an increased the risk for false associations. As a result, large prospective multi-institutional studies examining PNS resolution in metastatic RCC patients are needed to validate our findings. Further research is needed to determine how best to incorporate PNS resolution into current prognostic models. 

## 5. Conclusions

Among patients with mRCC, those who did not have resolution of their PNS within one year had worse OS and CSS compared to patients with PNS resolution. While scoring poorly on current mRCC models such as SCREEN, IMDC, and MSKCC was associated with worse OS, there was little correlation with PNS resolution. Thus, PNS resolution may offer distinct prognostic information not captured in current mRCC models. 

## Figures and Tables

**Figure 1 cancers-16-03678-f001:**
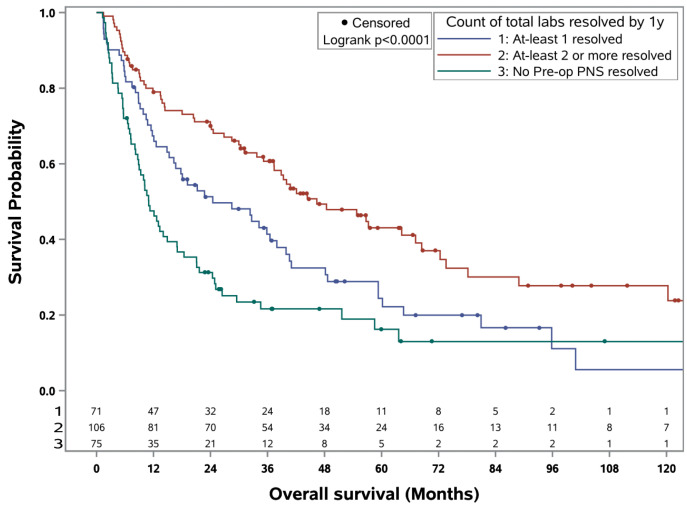
Kaplan–Meier curve for overall survival in patients with RCC by count of paraneoplastic labs resolved by one year.

**Figure 2 cancers-16-03678-f002:**
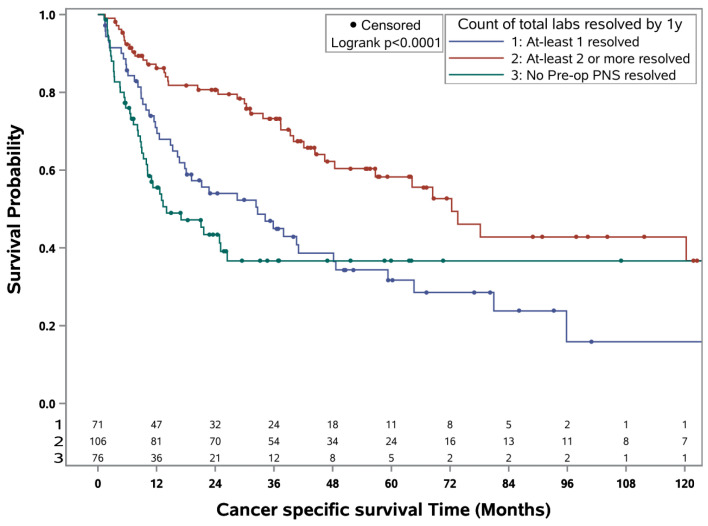
Kaplan–Meier curve for cancer-specific survival in patients with RCC by count of paraneoplastic labs resolved by 1 year.

**Table 1 cancers-16-03678-t001:** Distribution of cohort preoperative paraneoplastic abnormalities.

Preoperative PNS Labs	No	Yes	Missing
Low Hemoglobin	89 (33.7)	175 (66.3)	0
Low Hematocrit	122 (46.2)	142 (53.8)	0
Polycythemia	263 (99.6)	1 (0.4)	0
Elevated CRP	26 (11.6)	198 (88.4)	40
Elevated ESR	46 (22.9)	155 (77.1)	63
Hepatic Dysfunction	158 (61.5)	99 (38.5)	7
Corrected Hypercalcemia	218 (82.9)	45 (17.1)	1
Hyperuricemia	8 (72.7)	3 (27.3)	253
Thrombocytopenia	258 (97.7)	6 (2.3)	0
Thrombocytosis	231 (87.5)	33 (12.5)	0
High NLR	104 (49.1)	108 (50.9)	52

Abbreviations: C-reactive protein (CRP), erythrocyte sedimentation rate (ESR), and neutrophil–lymphocyte ratio (NLR).

**Table 2 cancers-16-03678-t002:** Distribution of cohort patient and surgical characteristics.

		Resolution of At Least One PNS 30 d–1 y Following Surgery			
Covariant		No N = 76	Yes N = 177	Total N = 253	*p*-Value
Age > 60		25 (32.9)	80 (45.2)	105 (41.5)	0.069
Male		59 (77.6)	118 (66.7)	177 (69.96)	0.081
Race					0.91
	Black	12 (15.8)	31 (17.5)	43 (17)	
	Other	6 (7.9)	12 (6.8)	18 (7.11)	
	White	58 (76.3)	134 (75.7)	192 (75.89)	
History of smoking		45 (59.2)	92 (52)	137 (54.15)	0.29
ECOG ≥ 1		20 (26.3)	43 (24.3)	63 (24.9)	0.733
BMI ≥ 30		25 (32.9)	68 (38.4)	93 (36.76)	0.404
Diabetes		21 (27.6)	54 (30.5)	75 (29.64)	0.646
HTN		47 (61.8)	118 (66.7)	165 (65.22)	0.46
CCI 4+		62 (81.6)	153 (86.4)	215 (84.98)	0.321
ccRCC		57 (75)	141 (79.7)	198 (78.26)	0.41
pT stage					0.288
	T1–T2	8 (10.5)	14 (7.9)	22 (8.7)	
	T3	63 (82.9)	140 (79.1)	203 (80.24)	
	T4	5 (6.6)	23 (13)	28 (11.07)	
IVC thrombus present		21 (27.6)	84 (47.5)	105 (41.5)	0.003
Necrosis		62 (81.6)	142 (80.2)	204 (80.63)	0.803
Nodal involvement					0.536
	pN0	22 (28.9)	64 (36.2)	86 (33.99)	
	pN1	32 (42.1)	68 (38.4)	100 (39.53)	
	pNx	22 (28.9)	45 (25.4)	67 (26.48)	
Tumor size *		9.6 (7–11.5)	11.2 (8.4–14)	10.5 (8–13.2)	<0.001
Receipt of adjuvant or salvage therapy		46 (60.5)	123 (69.5)	169 (66.8)	0.165
SCREEN risk category					0.376
	Favorable	10 (18.5)	16 (11.3)	26 (13.27)	
	Intermediate	25 (46.3)	67 (47.2)	92 (46.94)	
	Poor	19 (35.2)	59 (41.5)	78 (39.8)	
MKSCC risk score (score 2+)					0.083
	Favorable-intermediate	12 (24.5)	17 (13.6)	29 (16.67)	
	Poor	37 (75.5)	108 (86.4)	145 (83.33)	
IMDC risk score (score 2+)					0.029
	Favorable-intermediate	47 (61.8)	83 (46.9)	130 (51.38)	
	Poor	29 (38.2)	94 (53.1)	123 (48.62)	

* Median (IQR). Parametric *p*-value by ANOVA for numerical and chi-square test for categorical covariates. Numbers might not add up due to missing data. Abbreviations: paraneoplastic syndrome (PNS), ECOG Status (Eastern Cooperative Oncology Group Status), Body Mass Index (BMI), hypertension (HTN), Charlson comorbidity index (CCI), clear cell renal cell carcinoma (ccRCC), inferior vena cava (IVC), the Selection for Cytoreductive Nephrectomy (SCREEN), International Metastatic RCC Database Consortium (IMDC), Memorial Sloan-Kettering Cancer Center (MSKCC).

**Table 3 cancers-16-03678-t003:** Association of paraneoplastic resolution at 1 year in metastatic RCC patients with SCREEN, IMDC, and MSKCC scoring.

		SCREEN Risk			IMDC Risk *			MSKCC Risk *		
Lab Resolved by 1 Year (Threshold)	n(%)	Favorable-	Poor (N = 79)	*p*-Value	Favorable-	Poor (N = 116)	*p*-Value	Favorable-	Poor (N = 135)	*p*-Value
		Intermediate			Intermediate			Intermediate		
		(N = 125)			(N = 55)			(N = 29)		
Anemia resolution										
(<11 g/dL)	Yes	46 (67.6)	31 (56.4)	0.198	18 (58.1)	46 (60.5)	0.814	7 (77.8)	53 (57.6)	0.24
	No	22 (32.4)	24 (43.6)		13 (41.9)	30 (39.5)		2 (22.2)	39 (42.4)	
Low hematocrit resolution										
(<33%)	Yes	39 (75)	25 (52.1)	0.017	14 (60.9)	38 (59.4)	0.9	7 (87.5)	43 (56.6)	0.09
	No	13 (25)	23 (47.9)		9 (39.1)	26 (40.6)		1 (12.5)	33 (43.4)	
Thrombocytosis resolution										
(platelet count > 400/nL)	Yes	5 (50)	8 (50)	1	1 (25)	11 (50)	0.356	0 ()	12 (50)	0.09
	No	5 (50)	8 (50)		3 (75)	11 (50)		0 ()	12 (50)	
Hypercalcemia										
(corrected calcium >10.4 mg/dL)	Yes	10 (58.8)	13 (72.2)	0.404	5 (71.4)	16 (76.2)	0.801	0 ()	19 (76)	0.24
	No	7 (41.2)	5 (27.8)		2 (28.6)	5 (23.8)		0 ()	6 (24)	
High CRP resolution										
(>5 mg/L)	Yes	33 (37.9)	12 (24.5)	0.11	9 (23.7)	27 (33.8)	0.267	6 (35.3)	29 (30.9)	0.717
	No	54 (62.1)	37 (75.5)		29 (76.3)	53 (66.3)		11 (64.7)	65 (69.1)	
Elevated ESR resolution										
(M: >22 mm/h, F: >29 mm/h)	Yes	17 (39.5)	11 (39.3)	0.983	5 (26.3)	16 (40)	0.305	2 (28.6)	18 (38.3)	0.619
	No	26 (60.5)	17 (60.7)		14 (73.7)	24 (60)		5 (71.4)	29 (61.7)	
High LFT resolution										
(ALT > 50 U/L, AST > 50 U/L)	Yes	23 (46.9)	9 (27.3)	0.073	8 (50)	16 (36.4)	0.34	6 (50)	17 (36.2)	0.381
	No	26 (53.1)	24 (72.7)		8 (50)	28 (63.6)		6 (50)	30 (63.8)	
High NLR resolution										
(>4:1 ratio)	Yes	18 (37.5)	21 (42)	0.649	6 (54.5)	25 (43.9)	0.515	3 (30)	27 (47.4)	0.308
	No	30 (62.5)	29 (58)		5 (45.5)	32 (56.1)		7 (70)	30 (52.6)	

* A score of 2+ was used to indicate poor risk. Parametric *p*-value calculated by chi-square test. Numbers might not add up due to missing data. Abbreviations: The Selection for Cytoreductive Nephrectomy (SCREEN), International Metastatic RCC Database Consortium (IMDC), Memorial Sloan-Kettering Cancer Center (MSKCC), C-reactive protein (CRP), erythrocyte sedimentation rate (ESR), liver function test (LFT), and neutrophil–lymphocyte ratio (NLR).

**Table 4 cancers-16-03678-t004:** Multivariable Cox Hazards model for cancer-specific and overall survival.

		Overall Survival			Cancer-Specific Survival	
Covariant		n (%)	Hazard Ratio (95% CI)	*p*-Value	Hazard Ratio (95% CI)	*p*-Value
Count of total preop labs resolved at 1 year						
	≥1 lab resolved	71 (28.1)	1.76 (1.20–2.59)	0.004	2.24 (1.43–3.51)	<0.001
	No labs resolved	76 (30)	2.75 (1.86–4.07)	<0.001	2.62 (1.64–4.19)	<0.001
	≥2 labs resolved	106 (41.9)	Ref		Ref	
Receipt of systemic adjuvant treatment		143 (54.2)	1.31 (0.94–1.81)	0.108	1.37 (0.93–2.02)	0.108
Age > 60		143 (54.2)	1.06 (0.76–1.47)	0.744	0.82 (0.56–1.20)	0.31
Male		187 (70.8)	1.05 (0.75–1.47)	0.778	0.99 (0.67–1.46)	0.957
Race						
	Black	46 (17.4)	1.12 (0.74–1.69)	0.584	1.14 (0.71–1.83)	0.589
	Other	18 (6.8)	0.96 (0.52–1.78)	0.9	0.83 (0.39–1.77)	0.639
	White	200 (75.8)	Ref		Ref	
ECOG ≥ 1		68 (25.8)	1.66 (1.17–2.36)	0.005	2.13 (1.42–3.19)	<0.001
BMI ≥ 30 kg/m^2^		68 (25.8)	0.73 (0.52–1.02)	0.063	0.60 (0.40–0.90)	0.014
CCI 4+		68 (25.8)	0.61 (0.40–0.95)	0.028	0.46 (0.28–0.74)	0.001
pT stage						
	T3	212 (80.3)	1.29 (0.70–2.39)	0.418	1.57 (0.73–3.35)	0.249
	T4	28 (10.6)	1.77 (0.83–3.78)	0.143	2.19 (0.88–5.41)	0.09
	T1-T2	24 (9.1)	Ref		Ref	
IVC thrombus present		109 (41.3)	1.12 (0.80–1.57)	0.508	0.90 (0.60–1.35)	0.608
ccRCC		205 (77.7)	0.93 (0.63–1.36)	0.695	0.81 (0.53–1.26)	0.352
Nodal involvement						
	pN1	104 (39.4)	1.37 (0.95–1.99)	0.09	1.54 (1.00–2.38)	0.052
	pNx	72 (27.3)	1.08 (0.70–1.67)	0.718	1.05 (0.62–1.77)	0.87
	pN0	88 (33.3)	Ref		Ref	
Tumor size **			1.04 (1.00–1.09)	0.055	1.04 (0.99–1.09)	0.141
SCREEN risk *						
	Intermediate risk	99 (48.5)	2.04 (1.01–4.10)	0.046		
	Poor risk	79 (38.7)	2.99 (1.45–6.14)	0.003		
	Favorable risk	26 (12.7)	Ref			
IMDC risk *						
	Poor risk	116 (67.8)	1.56 (1.01–2.41)	0.043		
	Favorable-Intermediate risk	55 (32.2)	Ref			
MSKCC risk *						
	Poor risk	135 (82.3)	1.78 (1.05–3.01)	0.031		
	Favorable-Intermediate risk	29 (17.7)	Ref			

* Individual multivariable analyses were performed adjusting for all the variables above except for count of total PNS labs resolved. A total of 204 observations were used of 204 available in the SCREEN data set with a Harrell’s concordance statistic estimate of 0.6475. A total of 170 observations were used of 171 available in the IMDC data set with a Harrell’s concordance statistic estimate of 0.6299. A total of 163 observations were used of 164 available in the MSKCC data set with a Harrell’s concordance statistic estimate of 0.6344. ** Tumor size was treated as a continuous variable. Abbreviations: ECOG Status (Eastern Cooperative Oncology Group Status), Body Mass Index (BMI), Charlson comorbidity index (CCI), clear cell renal cell carcinoma (ccRCC), the Selection for Cytoreductive Nephrectomy (SCREEN), International Metastatic RCC Database Consortium (IMDC), Memorial Sloan-Kettering Cancer Center (MSKCC).

**Table 5 cancers-16-03678-t005:** Univariable Cox hazards model for overall and cancer-specific survival for individual paraneoplastic lab resolution at one year.

		Overall Survival		Cancer-Specific Survival	
PNS Labs by 1 Year	n(%)	Hazard Ratio (95% CI)	*p*-Value	Hazard Ratio (95% CI)	*p*-Value
No anemia resolution	69 (41.6)	3.10 (2.12–4.56)	<0.001	3.34 (2.11–5.28)	<0.001
No low hematocrit resolution	55 (41.4)	2.53 (1.68–3.80)	<0.001	3.14 (1.94–5.08)	<0.001
No thrombocytosis resolution	15 (44.1)	0.82 (0.37–1.84)	0.631	0.80 (0.34–1.86)	0.604
No hypercalcemia resolution	15 (36.6)	3.98 (1.90–8.35)	<0.001	5.16 (2.10–12.69)	<0.001
No high CRP resolution	118 (69.0)	2.11 (1.38–3.23)	<0.001	2.58 (1.51–4.41)	<0.001
No elevated ESR resolution	58 (64.4)	3.01 (1.69–5.36)	<0.001	3.31 (1.63–6.70)	<0.001
No high LFT resolution	60 (63.2)	1.92 (1.14–3.22)	0.014	2.12 (1.14–3.95)	0.018
No high NLR resolution	60 (58.8)	1.80 (1.11–2.90)	0.016	1.52 (0.88–2.62)	0.13

Abbreviations: C-reactive protein (CRP), erythrocyte sedimentation rate (ESR), liver function test (LFT), and neutrophil–lymphocyte ratio (NLR).

## Data Availability

Individual participant data that underlie the results reported in this article, after deidentification, can be made available upon reasonable request by investigators who provide a methodological sound proposal following publication. Proposals should be directed to the corresponding author. To gain access, data requestors will need to sign a data access agreement.

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
