# Peer review of "Paraneoplastic Resolution Holds Prognostic Utility in Patients with Metastatic Renal Cell Carcinoma"

_cancers, 2024, doi:10.3390/cancers16213678_

Round 1

Reviewer 1 Report

Comments and Suggestions for Authors

The manuscript, entitled "Paraneoplastic Resolution Holds Prognostic Utility in Patients With Metastatic Renal Cell Carcinoma," by Palmateer et al., is well worth printing. In the manuscript, the authors document important findings on the prognostic significance of paraneoplastic syndrome (PNS) resolution following cytoreductive nephrectomy in patients with metastatic renal cell carcinoma (mRCC). Based on the data collected from 253 patients, the authors exhaustively examined how resolving PNS abnormalities within one year of surgery correlated with improved overall survival (OS) and cancer-specific survival (CSS) using a variety of pre-defined factors (Tables 1-3). The study highlights that the prognostic utility of PNS resolution in mRCC could offer insights beyond existing risk models, making it a potential postoperative care tool for mRCC patients. Hopefully, in the future, the authors could validate the current model via multi-institutional studies, incorporating PNS resolution as part of routine clinical practice for mRCC patients.    

Author Response

The manuscript, entitled "Paraneoplastic Resolution Holds Prognostic Utility in Patients With Metastatic Renal Cell Carcinoma," by Palmateer et al., is well worth printing. In the manuscript, the authors document important findings on the prognostic significance of paraneoplastic syndrome (PNS) resolution following cytoreductive nephrectomy in patients with metastatic renal cell carcinoma (mRCC). Based on the data collected from 253 patients, the authors exhaustively examined how resolving PNS abnormalities within one year of surgery correlated with improved overall survival (OS) and cancer-specific survival (CSS) using a variety of pre-defined factors (Tables 1-3). The study highlights that the prognostic utility of PNS resolution in mRCC could offer insights beyond existing risk models, making it a potential postoperative care tool for mRCC patients. Hopefully, in the future, the authors could validate the current model via multi-institutional studies, incorporating PNS resolution as part of routine clinical practice for mRCC patients.    

Thank you for your comments. We are hoping to validate our findings in collaboration with other institutions in the near future.

Reviewer 2 Report

Comments and Suggestions for Authors

The authors have carried out a retrospective study that involved evaluating cases meeting the criteria of paraneoplastic syndrome amongst those with renal cell carcinoma for 22 years. The study is very well carried out and the findings are very relevant clinically. However, the following points are for authors' considerations:

1.  Please justify how anemia can be considered as a PNS? Isn't it part of the RCC-related manifestation?

2. Has the cut-off values for each of the laboratory parameters considered in this study have not changed over the last 2 decades? I am sure the methods/equipments used for estimating the laboratory parameters could have been different at different time periods and would this be a limitation to be mentioned?

3. Please specify if a formal sample size estimation carried out in the statistical analysis section.

4. If sample size was not estimated, could you please provide a post-hoc power calculation in the discussion section? This might provide readers on the possible errors while interpreting the key findings from this manuscript.

5. Please use Bonferroni correction measures to adjust for applying multiple statistical analyses. Otherwise, there is a very high inflation of p-values.

6. What is the justification to use composite parameters for defining PNS and then considering only 'resolution of atleast one PNS' in Table 2? I am unable to see the relationship between various parameters included in defining PNS such as hemoglobin, CRP, ESR, platelet count, hepatic functions, etc. Hence, it would be more scientific and meaningful if separate analyses were carried out for each of the relevant outcomes such as you can combine hemoglobin related parameters together (low hemoglobin, low hematocrit and polycythemia), CRP and ESR together, etc. In this way, there is a background logic in analyzing the factors associate with each group which by themselves are homogenous.

7. Please adhere to the reporting guidelines for retrospective studies from EQUATOR network.

8. Provide a filled-in checklist for the reporting guidelines pertaining to your manuscript.

Author Response

  1. Please justify how anemia can be considered as a PNS? Isn't it part of the RCC-related manifestation?

Thank you for the fantastic question. While anemia is indeed a common manifestation of RCC, it is considered a paraneoplastic syndrome because it is indirectly caused by the systemic effects induced by the tumor. RCC is known to release cytokines such as IL-6, TNF-alpha, and IFN-gamma which interfere with erythropoiesis and iron regulation. Additionally, RCC is understood to disrupt erythropoietin and hypoxia-inducible factor (HIF) pathways further contributing to anemia. These systemic effects can also be observed in other solid malignancies of the breast, pancreas, and thyroid.       

  1. Has the cut-off values for each of the laboratory parameters considered in this study have not changed over the last 2 decades? I am sure the methods/equipments used for estimating the laboratory parameters could have been different at different time periods and would this be a limitation to be mentioned?

Thank you for your question. Cut-off values have certainly changed over the last 2 decades in response to new scientific evidence. We are unaware if the methods/equipment for estimating laboratory parameters have changed in that time. To acknowledge this potential limitation, we have added a sentence in page 12, line 288-299.

  1. Please specify if a formal sample size estimation carried out in the statistical analysis section.

Thank you for the comment. No formal sample size and power estimation was carried out considering this is the analysis of retrospective data. All patients meeting the inclusion criteria were included in this analysis.

  1. If sample size was not estimated, could you please provide a post-hoc power calculation in the discussion section? This might provide readers on the possible errors while interpreting the key findings from this manuscript.

Thank you for the suggestion. In order to calculate post-hoc power with available population, it’s important to have a known estimate for resolution rate among metastatic RCC patients after nephrectomy. Since this study is the first of its kind to define and show the association of resolution rate with survival outcomes, the post-hoc power analysis is difficult to estimate. We hope to continue the enrollment for eligible patients and include power calculations in validation study planned in future.

  1. Please use Bonferroni correction measures to adjust for applying multiple statistical analyses. Otherwise, there is a very high inflation of p-values.

We appreciate your suggestion and agree that applying multiple statistical analyses can inflate p-values and produce false positives. Bonferroni correction, while it reduces the probability of false positives, it also increases the likelihood of false negatives. Given this is the first study of its kind, we felt that minimizing the risk of missing true associations was of greater importance. We included an acknowledgement of an increased risk of false positives within the limitations on page 12, lines 297-299.

  1. What is the justification to use composite parameters for defining PNS and then considering only 'resolution of atleast one PNS' in Table 2? I am unable to see the relationship between various parameters included in defining PNS such as hemoglobin, CRP, ESR, platelet count, hepatic functions, etc. Hence, it would be more scientific and meaningful if separate analyses were carried out for each of the relevant outcomes such as you can combine hemoglobin related parameters together (low hemoglobin, low hematocrit and polycythemia), CRP and ESR together, etc. In this way, there is a background logic in analyzing the factors associate with each group which by themselves are homogenous.

Thank you for the suggestion. This is a fantastic idea and one which we had initially considered. The aim of our study has been to make an initial examination of whether the presence and number of paraneoplastic syndromes resolved by one year was prognostic in patients with RCC. With only 253 patients in our cohort, we are afraid our study may be underpowered to evaluate these associations if we further subdivided patients. This would be an excellent project opportunity as a multi-institutional validation study and one we plan to do in the near future.  

  1. Please adhere to the reporting guidelines for retrospective studies from EQUATOR network.

Thank you for your recommendation. We have made the following changes in an effort to more closely adhere to the STROBE guidelines:

  1. For item 12c, we added a sentence on page 4, lines 116-118 addressing how missing data was handled.  
  2. For item 13c, we added a flow diagram as a supplementary figure and referenced it on page 3, line 82.
  3. For item 14c, we added follow-up data for the cohort on page 4, lines 134-135.

  1. Provide a filled-in checklist for the reporting guidelines pertaining to your manuscript.

Thank you for your suggestion. Please see the attached STROBE checklist.

Reviewer 3 Report

Comments and Suggestions for Authors

Journal: Cancers

Manuscript ID: cancers-3257286

Title: Paraneoplastic resolution holds prognostic utility in patients with metastatic renal cell carcinoma

Authors: Gregory S. Palmateer, Edouard H. Nicaise, Taylor Goodstein, Benjamin N. Schmeusser, Dattatraya Patil, Nahar Imtiaz, Daniel D. Shapiro, E. Jason Abel, Shreyas Joshi, Vikram Narayan, Kenneth Ogan, Viraj A.

The manuscript “Paraneoplastic resolution holds prognostic utility in patients with metastatic renal cell carcinoma” by Gregory S. Palmateer et. alis a great clinical cancer research article in a current and interesting topic and could be of interest for medical scientists. In this study, the authors reviewed a prospectively maintained nephrectomy database that included patients with metastatic renal cell carcinoma (mRCC) of any histology who underwent nephrectomy between 2000 and 2022. The manuscript is very well written, the methodology and results clearly described and discussed. Clinical study results show that resolution of preoperative PNS abnormalities within one year after surgery is associated with improved OS and CSS in patients with mRCC. This clinical research may contribute to precision medicine and cancer therapy. In my opinion, there are a couple minor issues that would affect the publication of this research article in its current form.

1.      Page 7, Table 3: the p-value format of Association of paraneoplastic resolution at 1 year in metastatic RCC patients with SCREEN, IMDC, and MSKCC scoring is not consistent.

2.      I strongly recommend that the authors reformat Tables 1, 2, and 4 to make it easier and clearer for readers to read

3.      This clinical research article is acceptable after minor revisions.

Author Response

The manuscript “Paraneoplastic resolution holds prognostic utility in patients with metastatic renal cell carcinoma” by Gregory S. Palmateer et. al, is a great clinical cancer research article in a current and interesting topic and could be of interest for medical scientists. In this study, the authors reviewed a prospectively maintained nephrectomy database that included patients with metastatic renal cell carcinoma (mRCC) of any histology who underwent nephrectomy between 2000 and 2022. The manuscript is very well written, the methodology and results clearly described and discussed. Clinical study results show that resolution of preoperative PNS abnormalities within one year after surgery is associated with improved OS and CSS in patients with mRCC. This clinical research may contribute to precision medicine and cancer therapy. In my opinion, there are a couple minor issues that would affect the publication of this research article in its current form.

  1. Page 7, Table 3: the p-value format of Association of paraneoplastic resolution at 1 year in metastatic RCC patients with SCREEN, IMDC, and MSKCC scoring is not consistent.

Thank you for bringing this to our attention. We have taken your suggestion and have corrected the p-value format to be consistent across Table 3.

  1. I strongly recommend that the authors reformat Tables 1, 2, and 4 to make it easier and clearer for readers to read

Thank you for your feedback. We have reformatted Tables 1,2, and 4 and have adjusted the remaining tables to share similar formatting in an attempt to make it easier for readers to read.

  1. This clinical research article is acceptable after minor revisions.

Thank you for your insightful and detailed comments. 

Round 2

Reviewer 2 Report

Comments and Suggestions for Authors

Thank you for the revision.